# Competitiveness during Dual-Species Biofilm Formation of *Fusarium oxysporum* and *Candida albicans* and a Novel Treatment Strategy

**DOI:** 10.3390/pharmaceutics14061167

**Published:** 2022-05-30

**Authors:** Annarita Falanga, Angela Maione, Alessandra La Pietra, Elisabetta de Alteriis, Stefania Vitale, Rosa Bellavita, Rosa Carotenuto, David Turrà, Stefania Galdiero, Emilia Galdiero, Marco Guida

**Affiliations:** 1Department of Agricultural Science, University of Naples “Federico II”, Via Università 100, 80055 Portici, Italy; annarita.falanga@unina.it (A.F.); davturra@unina.it (D.T.); 2Department of Biology, University of Naples ‘Federico II’, Via Cinthia, 80126 Naples, Italy; angela.maione@unina.it (A.M.); ale.lapietra094@gmail.com (A.L.P.); dealteri@unina.it (E.d.A.); rosa.carotenuto@unina.it (R.C.); marco.guida@unina.it (M.G.); 3Institute for Sustainable Plant Protection, National Research Council of Italy, 80055 Portici, Italy; stefania.vitale@ipsp.cnr.it; 4Department of Pharmacy, School of Medicine, University of Naples “Federico II”, Via Domenico Montesano 49, 80131 Naples, Italy; rosa.bellavita@unina.it (R.B.); sgaldier@unina.it (S.G.); 5Center for Studies on Bioinspired Agro-Enviromental Technology, Università di Napoli “Federico II”, 80055 Portici, Italy

**Keywords:** *Fusarium oxysporum*, *Candida albicans*, polymicrobial biofilm, WMR, *Galleria mellonella*

## Abstract

During an infection, a single or multispecies biofilm can develop. Infections caused by non-dermatophyte molds, such as *Fusarium* spp. and yeasts, such as *Candida* spp., are particularly difficult to treat due to the formation of a mixed biofilm of the two species. *Fusarium oxysporum* is responsible for approximately 20% of human fusariosis, while *Candida albicans* is responsible for superficial mucosal and dermal infections and for disseminated bloodstream infections with a mortality rate above 40%. This study aims to investigate the interactions between *C. albicans* and *F. oxysporum* dual-species biofilm, considering variable formation conditions. Further, the ability of the WMR peptide, a modified version of myxinidin, to eradicate the mixed biofilm when used alone or in combination with fluconazole (FLC) was tested, and the efficacy of the combination of WMR and FLC at low doses was assessed, as well as its effect on the expression of some biofilm-related adhesin and hyphal regulatory genes. Finally, in order to confirm our findings in vivo and explore the synergistic effect of the two drugs, we utilized the *Galleria mellonella* infection model. We concluded that *C. albicans* negatively affects *F. oxysporum* growth in mixed biofilms. Combinatorial treatment by WMR and FLC significantly reduced the biomass and viability of both species in mature mixed biofilms, and these effects coincided with the reduced expression of biofilm-related genes in both fungi. Our results were confirmed in vivo since the synergistic antifungal activity of WMR and FLC increased the survival of infected larvae and reduced tissue invasion. These findings highlight the importance of drug combinations as an alternative treatment for *C. albicans* and *F. oxysporum* mixed biofilms.

## 1. Introduction

Filamentous fungi belonging to *Fusarium* spp. are ubiquitously present in nature, and their spores are usually dispersed into the environment through water, wind, and additional abiotic or biotic vectors. The genus *Fusarium* comprises a large set of soil-dwelling and plant interacting species that include parasites, endophytes, and biocontrol agents. Aside from this, *Fusarium* spp. are increasingly being reported as opportunistic pathogens of invertebrates and mammals, with several isolates able to efficiently cause disease on both plants and animals [1,2].

Because of this unusual ability of *Fusarium* to undergo host jumps that cross kingdom of life barriers, *Fusarium oxysporum* has been previously suggested as a model organism to study the molecular basis of disease in cross-kingdom pathogens [3]. Human infections caused by *Fusarium* spp. are increasingly being reported by the scientific community as a consequence of more reliable techniques to identify fungal isolates in clinical samples and of the growing number of immunocompromised patients. In human patients, *Fusarium* spp. are the cause of a wide spectrum of infections that include mycetoma, keratitis, and onychomycosis in immunocompetent individuals and locally invasive or systemic infections in immunocompromised ones, often leading to lethal outcomes. Fusariosis, the invasive fungal infection caused by *Fusarium* spp., is currently listed as the second most common opportunistic human infection in the world after aspergillosis. *Fusarium* dissemination and infection mainly occur through airborne transmission, trauma-driven skin breakdown due to trauma, burns, or insertion of vascular catheters [4,5].

Importantly, *Fusarium* species are intrinsically resistant to most common commercial antifungals such as triazoles and echinocandins and able to produce 3D biofilm structures, which are several folds more resistant than planktonic cells to stressing agents and targeting compounds [6,7].

Given that 60–80% of human infections are estimated to be biofilm-related and that *Fusarium* spp. are highly resistant to antifungal treatments, new therapies for the treatment of fusariosis are urgently needed [8].

*C. albicans* is the pathogen most commonly isolated from superficial mycoses or causing systemic and invasive infections. Predisposing circumstances such as extreme age, reduced immune system, diabetes mellitus, HIV/AIDS, and uncontrolled use of broad-spectrum antibiotics favor the establishment of candidiasis. One of the major virulence factors of *Candida* is its ability to form biofilms, which has also been widely reported in host tissues, catheters, and indwelling medical devices [9].

Biofilms consist of communities of cells embedded in a polymeric extracellular matrix and can be composed of microorganisms of different species. This condition allows sessile microbial cells to perform various functions, including nutrition, excretion, growth, communication, and protection, with high efficiency. Communities formed by fungi are made up of a dense network of interconnected hyphae covered and embedded by an extracellular matrix that can be dense or thin, and changes in morphology are often regulated by molecules used for communication between cells through a mechanism known as quorum-sensing [10].

The study of polymicrobial biofilms has been increasing, and, in the medical area, the role played by biofilms in co-infections has been considered a determinant virulence factor for pathogenesis in the host [11].

Clinical practice cases and unpublished studies have demonstrated that it is possible to isolate one or more microorganisms from several infection sites in mycoses. Although the formation of mixed biofilms between molds such as *Fusarium* spp. and yeasts of the genus *Candida* has not been demonstrated so far, its occurrence is likely, especially in immunocompromised patients, since both species are able to form biofilms on biotic and abiotic surfaces [12].

Furthermore, in clinical practice, the unsatisfactory use of monotherapy in the treatment of fusariosis, the lack of efficacy of antifungals available for candidemia due to the development of biofilms, and their toxicity have led to the study of alternative treatment strategies. Particularly, the association of pre-existing antifungal drugs and new antimicrobial peptides has become a promising alternative for combating fungal biofilms [13,14]. The therapeutic strategy of combining the use of antimicrobial peptides (AMPs) and conventional therapeutic agents allows dose reduction, leading to minor toxicity, the attenuation of side effects, and increasing selectivity [15].

Therefore, AMPs have been considered a promising treatment option to combat the increasing number of drug-resistant pathogens. The WMR peptide, a modified version of the myxinidin sequence developed in our laboratory, has shown a high antimicrobial activity not only against Gram-positive and Gram-negative bacteria [16] but also antifungal activity against several *Candida* species [17]. WMR displays a versatile mechanism of action that varies from cell membrane damage to interactions with intracellular targets, and it is also effective towards biofilms.

Here, we report the ability of *Fusarium oxysporum* human isolate, NRRL 32931, and *Candida albicans* ATCC 90028 to form a dual-species biofilm, investigating the variable conditions of formation and evaluating the effects of the antimicrobial peptide WMR alone and in combination with fluconazole (FLC) in removing the mature biofilms of the two fungal species. Furthermore, to validate our findings in vivo, we used the non-vertebrate model host *Galleria mellonella* larvae infected with microconidia of *F. oxysporum* and *C. albicans* cells and tested the effectiveness of WMR and FLC alone or in combination on survival and tissue invasion.

## 2. Materials and Methods

### 2.1. Strains, Media, and Growth Conditions

The strains used in this study were *Candida albicans* ATCC 90028 and *Fusarium oxysporum* NRRL 32931 [18] isolated from the blood of a leukemia patient with invasive fusariosis, provided by the Department of Agricultural Science, University of Naples “Federico II”. *Candida albicans* ATCC 90028 was maintained on Sabouraud dextrose agar (OXOID, Basingstoke, UK) and grown in Tryptic Soy Broth (TSB) (OXOID, Basingstoke, UK) with 1% glucose for 18 h under shaking conditions (170 rpm) at 37 °C. Then, cells were centrifuged at 3000× *g* for 5 min at 4 °C, washed twice with phosphate-buffered saline (PBS) (OXOID, Basingstoke, UK), and standardized to 10^6^ cells mL^−1^ for next use. *Fusarium oxysporum* was grown on Potato Dextrose Agar (OXOID, Basingstoke, UK) and incubated at 28 °C for 5 d. Conidia were obtained by inoculating small pieces of a fungal colony in Potato Dextrose Broth for 72 h at 28 °C. Then, the culture medium was filtered through sterile gauze to remove hyphae, centrifuged (10 min at 2700× *g*), and the pellet containing conidia was resuspended in PBS. Conidia were counted in a Bürker chamber. The suspension of conidia was diluted in PBS to obtain a conidial suspension of 2.5 × 10^6^ conidia mL^−1^ to be used for further experiments.

### 2.2. Peptide Synthesis

WMR (NH_2_-WGIRRILKYGKRSK-CONH_2_) was synthesized based on ultrasonic-assisted solid-phase peptide synthesis (US–SPPS) methodology) as previously reported [17]. Briefly, Fmoc deprotection (20% piperidine in dimethylformamide (DMF, Merk Life Science S.r.l., Milano, Italy), 0.5 + 1 min) and the coupling (HOBt/HBTU, purchased by Merk Life Science S.r.l., Milano, Italy as additive reagents (3 equiv), and DIPEA (purchased by Merk Life Science S.r.l., Milano, Italy 6 equiv) as basis, 2 × 10 min) reactions, were cyclically performed until the obtainment of peptide sequence. Rink amide MBHA resin (0.58 mmol/g) was used. The crude peptide was cleaved from the resin with an acid solution, TFA:TIS:H_2_O (95:2.5:2.5, *v*/*v*/*v* purchased by Merk Life Science S.r.l., Milano, Italy), precipitated with cold ethyl ether, and analyzed by ESI LC–MS and RP-HPLC using a gradient of acetonitrile (0.1% TFA) in water (0.1% TFA) from 10% to 70% in 15 min. The crude peptide was purified by preparative RP-HPLC and obtained with a good yield (70%).

### 2.3. Determination of Minimum Inhibitory Concentrations (MICs)

The two fungal species were evaluated for susceptibility to both the WMR peptide ranging from 12.5 to 150 μM and FLC ranging from 30 to 150 μM (corresponding to 9.2 to 46 μg mL^−1^). MICs were determined according to the Clinical and Laboratory Standards Institute (CLSI) broth microdilution method for filamentous fungi M38-A2 and yeast M27-A3 [19,20], respectively, and defined as the lowest concentration that showed 100% inhibition of visible growth compared to the drug-free control.

### 2.4. Biofilm Formation and Detection

Single biofilms of *C. albicans* and *F. oxysporum* and mixed *C. albicans/F. oxysporum* biofilms were formed on 96-well flat-bottom polystyrene plates. The mono-species biofilm of *C. albicans* was performed as reported in our previous study [21]. Briefly, *C. albicans* biofilm was allowed to form inoculating, at a final concentration of 10^6^ cells mL^−1^, 100 μL of RPMI per well in a microtiter plate, and incubating at a temperature of 37 °C for 48 h. Similarly, for the *F. oxysporum* biofilm [7], wells of a microtiter plate were inoculated with 100 μL each of RPMI containing 10^6^ conidia mL^−1^ and incubated for 48 h at 37 °C to allow the conidia to settle and adhere to the bottom of the plate.

For the dual-species biofilm, the suspensions of the two microorganisms were prepared at a final total concentration of 10^6^ cells mL^−1^ in RPMI (ratio 1:1), dispensed (100 µL per well), and incubated for 48 h at 37 °C, renewing the culture medium after 24 h.

Additional assays for mixed biofilm in order to analyze the behavior in vitro of *C. albicans* and *F. oxysporum* were developed at the following conditions: Mix A, *F. oxysporum* was seeded first for 4 h until adherence and was followed by the addition of *C. albicans*; Mix B, *F. oxysporum* was seeded first for 24 h followed by addition of *C. albicans*, and Mix C, *F. oxysporum* was seeded for 48 h followed by addition of *C. albicans*. For all three mixtures, polystyrene plates were incubated for 72 h at 37 °C, renewing the culture medium every 24 h.

The ability of the two strains to form mono- and dual-species biofilms was detected using the crystal violet staining method for total biofilm biomass determination, according to Stepanović et al. [22] and modified by Ramírez-Granillo et al. [23], which used crystal violet at 0.005% p/v (final concentration). Microplates were rinsed three times with PBS, fixed with 99% methanol for 15 min, and air-dried before adding 200 μL of 0.005% p/v crystal violet (Sigma-Aldrich, St. Louis, MO, USA) to each well. After incubation for 15 min at room temperature, wells were washed three times with PBS, and 300 μL of acetic acid (30% *v/v*) was added to each well. Plates were read at 570 nm using a plate reader (SYNERGY H4 BioTek, BioTek Instruments, Agilent Technologies, Winooski, VT 05404 USA).

Quantitative analysis of biofilm vital biomass was detected by XTT reduction assay by using the tetrazolium2,3-bis (2-methoxy-4-nitro-5 sulfophenyl)-5-[(phenylamine) carbonyl]-2H-hydroxide re-duction assay (XTT) (Sigma-Aldrich, St. Louis, MO, USA) according to the manufacturer’s instructions.

For enumerating colony-forming units (CFU) in biofilms, the plates were emptied and washed with PBS to remove non-adherent cells, and 100 μL of PBS was added to each well. Then, adherent cells were accurately scraped from wells, vigorously vortexed, and suspended in 1 mL of PBS. Decimal dilutions were prepared in PBS up to 10^−5^, and 100 μL suspension was plated on selective media: CHROM agar TM Candida (Difco, BD Biosciences) and modified Sabouraud with chloramphenicol and cycloheximide (VWR Chemicals) incubated at 37 °C for 48 h for *C. albicans*, and at 28 °C for 72 h for *F. oxysporum* detection, respectively.

### 2.5. Scanning Electron Microscopy (SEM) of Biofilms

To perform SEM observation of mixed biofilms, the biofilm-containing wells were washed with PBS, fixed overnight with 3% *v/v* glutaraldehyde at 4 °C, then washed with PBS and post-fixed in 1% p/v aqueous solution of osmium for 90 min at room temperature. Then, samples were dehydrated in a series of graded alcohols, dried to the critical drying point, and finally coated with gold, as reported previously [24]. Specimens were evaluated with a scanning electron microscope (QUANTA 200 ESEM FEI Europe Company, Eindhoven, The Netherlands).

### 2.6. Activity of WMR against Mono- and Dual-Species Biofilms

The activity of the peptide WMR was verified against mono- and dual-species biofilms formed using the previously described conditions. Mature biofilms, after 72 h, were treated with 50, 100, and 150 µM of WMR for another 24 h at 37 °C. The residual biomass after the eradication activity of WMR was quantified by crystal violet assay. The percentage of eradication was calculated as biofilm reduction% = (OD_570_ control − OD_570_ sample/OD_570_ control) × 100, where OD_570_ control and OD_570_ sample corresponded to OD_570_ of the untreated and treated biofilm, respectively [25].

### 2.7. Checkerboard Assays: Assessment of the In Vitro Synergy Activity of WMR in Combination with FLC

In order to verify whether the antifungal agents in combination had the effect of removing the formed biofilm, combinations between the antifungal agents were established. The synergistic activity was evaluated by calculation of the sessile fractional inhibitory concentration index (sFICI). Briefly, biofilm cells were treated with various combinations of test agents (30, 60, 120, and 240 µM FLC and 12.5, 25, 50, and 100 µM WMR) by adding 50 μL of each prepared dilution in the vertical and horizontal direction of the plate, respectively. Further, the plate was incubated at 37 °C for 24 h, and then the remaining contents were removed, washed, and assessed with detection of total biomass and CFU. sFICI for all the combinations was determined as [(minimum concentration of drug A in combination)/(minimum concentration of drug alone)]/[(minimum concentration of drug B in combination)/(minimum concentration of drug B alone)]. The sFICI result was interpreted as: synergistic: FICI ≤ 0.5, no interaction or indifferent > 0.5–4.0, antagonistic > 4.0 [26].

### 2.8. qRT-PCR

Expression of *ALS3* and *ERG11* genes for *C. albicans* and *VeA* and *VelB* for *F. oxysporum* was analyzed using qRT-PCR, as previously reported in mixed biofilms treated with WMR and FLC at the synergic concentration [27]. Biofilms washed with PBS were then scraped and collected in 1.5 mL centrifuge tubes. The samples were centrifuged, and the supernatant was discarded. The pellet was used to extract total RNA. For RNA extraction, Direct-zolTM RNA Miniprep Plus Kit (ZYMO RESEARCH, Irvine, CA, USA) was used, and the purity and concentration of the extracted RNA were verified using Nanodrop spectrophotometer 2000 (Thermo Scientific Inc., Waltham, MA, USA). For each sample, 1000 ng of total RNA was retrotranscribed with an iScript™ cDNA Synthesis kit (Bio-Rad, Milan, Italy), following the manufacturer’s instructions. Primers used for analysis were reported in Appendix A. Quantitative PCR was performed in an AriaMx Real-Time PCR instrument (Agilent Technologies, Inc., Santa Clara, CA, USA), according to the manufacturer’s instructions of the System thermal cycler. SYBR Green qPCR SuperMix (TransGen Biotech, Beijing, China) was used, and fluorescence was measured using Agilent Aria 1.7 software (Agilent Technologies, Inc.). Thermal cycler protocol consisted of 95 °C for 10 min, one cycle for cDNA denaturation; 95 °C for 15 s and 60 °C for 1 min, 40 cycles for amplification; 95 °C for 15 s, one cycle for final elongation; and one cycle for melting curve analysis (from 60 °C to 95 °C) to verify the presence of a single product. The expression of each gene was analyzed and normalized against the *ACT1* gene and *FOYG* gene using REST software (Relative Expression Software Tool, Weihenstephan, Germany, version 1.9.12) based on the Pfaffl method [28,29].

### 2.9. In Vivo Effect of WMR Alone and in Combination with FLC Using the G. mellonella Infection Model

Efficacy of WMR and FLC, alone or in combination, in *G. mellonella* larvae infected with *C. albicans*, *F. oxysporum*, or the mixture of the two species, was evaluated by survival assays, as described previously [17].

Groups of 20 larvae were maintained in wood shavings in the dark at room temperature before use. In all the experiments, untouched larvae and larvae injected with 10 μL sterile PBS were included as control. To evaluate in vivo toxicity of the two different drugs, we first carried out survival assays on *G. mellonella* larvae injected with WMR, FLC, and the two compounds at the synergistic concentration established in the checkboard assay (12.5 μM WMR and 30 μM FLC).

To test the efficacy of WMR and FLC or their combination, larvae were injected with each of the two pathogens: *F. oxysporum* conidia (10^6^ conidia per larva), *C. albicans* cells (10^6^ yeast cells per larva), or the mixture of conidia and Candida cells (10^6^ in total per larva) via the last right proleg using a Hamilton syringe. Then, an aliquot of 10 μL of 12.5 μM WMR or 1.63 μM FLC or mixed drugs was delivered behind the last proleg on the opposite side of the pathogen injection site 2 h post-infection for treatment experiments. All groups of larvae were incubated at 35 °C in the dark. Survival was recorded every 24 h for a duration of 4 days, and larvae were considered dead if they gave no response to slight touch. *G. mellonella* survival curves were analyzed by the Kaplan–Meier method. Differences between groups were considered significant at *p* < 0.05.

### 2.10. Histological Analyses

*Candida* and *Fusarium* presence alone or together within *G. mellonella* tissues was assessed via collecting larvae, injected with the pathogens, as described above, from each group on day 3 post-infection and treatment. For histology, the infected and control larvae were processed with histological protocols for optical microscopy. Briefly, the samples were fixed with 10% neutral formalin; then, they were dehydrated with an ethanol gradient (70, 80, 90, 96, and 100% ethanol) and embedded in paraffin. Hematoxylin–eosin staining was performed on sections of 10 µm [30]. The images were acquired with a Zeiss Axiocam Microscope Camera Applied to a Zeiss Axioscope microscope (Zeiss, Jena, Germany).

### 2.11. Statistical Analysis

The results were obtained from three independent experiments, and data were shown as mean values ± standard deviation (SD). Data were analyzed by GraphPad Prism Software (version 8.02 for Windows, GraphPad Software, La Jolla, CA, USA, www.graphpad.com accessed on 10 March 2022), using Tukey’s multiple comparison test following one-way ANOVA. Survival curves were plotted using the Kaplan–Meier method. Asterisks indicate significant differences (* = *p* < 0.05, ** = *p* < 0.01, *** = *p* < 0.001, **** = *p* < 0.0001).

## 3. Results

The susceptibility of *F. oxysporum* and *C. albicans* planktonic cells to antifungal agents is shown in Table 1. While WMR exhibited good antifungal activity against *C. albicans*, as also reported in our previous study [17], the complete growth inhibition of *F. oxysporum* was not found since MIC values were higher than 150 µM. All the two species indiscriminately showed high MIC values for fluconazole > 150 μM (corresponding to 46 μg mL^−1^).

The two strains tested displayed the capacity to form mono- and polymicrobial mature biofilms. Biofilm formation was measured at 48 h for single and mixed biofilms. As shown in Figure 1 (panel a), *F. oxysporum* was a strong biofilm producer, while *C. albicans* was an average producer of biofilm in vitro. The total biomass of the dual-species biofilm *C. albicans*/*F. oxysporum*, formed when the two species were dispended in the microplate at a ratio of 1:1, was significantly lower when compared to the single biofilm of *F. oxysporum* but superior to that of the single *C. albicans* biofilm (*p* < 0.0001, *p* < 0.01). Instead, the metabolic activities of the biofilms, determined by XTT reduction assay in Figure 1 (panel b), were similar.

The competitive behavior of *C. albicans* and *F. oxysporum* during in vitro mixed biofilm formation is clear and shown in Figure 2. When *C. albicans* was added after 4 or 24 h from *Fusarium* in the procedure of dual-species biofilm formation (Mix A and Mix B, respectively), the resulting vital biomass was lower than that observed in Mix C when *Candida* inoculum was added after 48 h from *Fusarium*, allowing the establishment and maturation of the biofilm of the filamentous fungus (Figure 2, panel a).

Figure 2, panel b shows the significant predominance of *C. albicans* compared to *F. oxysporum*: indeed, the CFU count depicted a remarkable reduction of the mold in all the three conditions tested. The predominance of *C. albicans* in comparison to *F. oxysporum* revealed the antagonistic interaction between the two species.

The co-existence of *C. albicans* and *F. oxysporum* is clearly visible in the SEM image of the dual-species biofilm (Figure 3). Indeed, it is characterized by a dense network of *C. albicans* yeast-like cells and elongated fungal hyphae, with a predominance of yeast-like cells.

Only one condition (Mix C) for the formation of dual-species biofilms was chosen as a model to determine the ability of WMR and FLC alone or in combination to eradicate mature mixed biofilm.

The minimal concentrations of WMR and FLC that eradicate mono- and dual-species pre-formed biofilms were monitored. As shown in Figure 4, the application of either WMR or FLC alone was not sufficient to completely eradicate mono- and dual-species biofilms of *F. oxysporum* within the concentration ranges tested, reaching 40% eradication at 150 μM concentration. On the contrary, we confirmed the good performance of WMR (Figure 4a) on *C. albicans* biofilm eradication.

The eradication of mixed biofilms of 50% was only observed at the highest WMR concentration tested.

A scarce activity of FLC against mono and dual-species of biofilm was observed in the case of FLC (Figure 4b): FLC was able to reduce *Candida* biofilm, but not above 55%.

The interaction between WMR and FLC in exerting antibiofilm activity against pre-formed dual-species biofilms formed by *C. albicans* and *F. oxysporum* was analyzed through the checkerboard assay. The synergistic effects of the two antifungal agents, when combined, as well as the ability of the association to eradicate mature biofilm, are shown in Figure 5. The heatmap (Figure 5a) displayed the synergistic antibiofilm efficacy of the interaction at the following combination of 12.5 µM and 30 µM for WMR and FLC, respectively, and the sFICI was found to be higher than 0.5. However, the application of WMR and FLC together caused a clear reduction in biofilm biomass in comparison to mono-treatment by each compound. The biomass of the dual-species biofilms was reduced by over 60% with combinatorial concentrations of the two compounds (Figure 5b). Additionally, viable biofilm cells in the presence of the synergistic combinations showed a drastic reduction in number (Figure 5c).

Transcriptional analysis of one of the ergosterol synthesis genes, *ERG11*, and one of the adhesin genes, *ALS3*, for *C. albicans* were investigated by qRT-PCR, as summarized in Figure 6. The expression levels of the two genes showed a differential behavior upon treatment of WMR/FLC. *ERG11* was moderately upregulated, whereas *ALS3* was significantly downregulated. *VeA* and *VelB* belong to the genes of the velvet regulation system, influencing the growth and development of the hypha, and conidiation as well as relations with the secondary metabolism for *F. oxysporum*, respectively. As expected, both *VeA* and *VelB* genes in the presence of WMR/FLC in synergistic combination were downregulated (1.59-fold, and 1.65-fold, respectively) (Figure 6).

The survival rate is the most important index to evaluate the effect of drugs in vivo with the *G. mellonella* infection model. To evaluate the toxicity of the different drugs, we first carried out survival assays on *G. mellonella* larvae injected with WMR, FLC, and the two compounds at the synergistic concentration of 12.5 μm and 30 μm, respectively. As shown in Figure 7, survival of 80% was found in larvae injected with WMR, FLC, and WMR plus FLC after 4 days of incubation. As depicted in Figure 8, after 4 days of incubation, groups of larvae infected with *F. oxysporum* and *C. albicans* alone or together had a survival between 20 and 30%, whereas survival increased to 60–70% with WMR or FLC and up to 80% when the two compounds were used at synergistic concentrations. This finding demonstrated that FLC plus WMR could significantly (*p* < 0.05, Tukey’s test) improve the survival rate of *G. mellonella* larvae when compared with FLC or WMR monotherapy.

In Figure 9, the results of the histological analysis performed on *G. mellonella* larvae infected with the two pathogens and treated with the combination WMR/FLC are reported. It is evident that in the tissues of treated animals, the presence of *C. albicans* cells and *F. oxysporum* hyphae was not detectable.

## 4. Discussion

Human-infecting pathogenic fungi are evolving resistance to all licensed systemic antifungal drugs. This increased incidence of drug resistance due to the implementation of antifungal drugs is an emerging problem that could be solved by alternative synergistic drug combinations among antifungals and new molecules in order to perfect innovative antifungal therapy against systemic and invasive mycoses [31,32,33].

More recently, some AMPs with known action against different microorganisms have been associated with conventional antimicrobial drugs to improve the action of these drugs and decrease microbial resistance [34,35].

The combinations evaluated here are unprecedented and are favorable for the eradication of *F. oxysporum*/*C. albicans* mixed biofilm.

Cells included in a biofilm form a heterogeneous and drug-tolerant natural environment that exhibits increased resistance to antifungal agents, mostly in mixed biofilm where heterogeneous species are embedded within a self-produced matrix consisting of extra polymeric substances. Thus, mixed fungal biofilms represent one of the main virulence factors contributing to the pathogenesis of fungal diseases

*Fusarium* spp. now represent the second most frequent mold causing invasive fungal infections after *Aspergillus* spp. and in particular, *F. oxysporum* causes infections in humans ranging from superficial dermato- or keratomycosis to fatal disseminated fusariosis with a clinical impact increased over the last decades [36].

*Candida* spp. [37] are the fungi that most frequently intervene in hospital infections, and *C. albicans* is an important opportunistic pathogen that is prevalent in clinical contexts, ranging from superficial mucosal and dermal infections to disseminated bloodstream infections with mortality rates above 40%.

Previously, WMR was found to be active against both *C. albicans* and non-*albicans* planktonic cells, and was also effective in both preventing the formation and eradicating *Candida* mature biofilms [17]. The present work confirms the good anti-*Candida* activity of WMR. Instead, *F. oxysporum* was found to be refractory to the peptide. Indeed, *F. oxysporum* is known to be very resistant to common antifungals [36].

In this study, we report the formation of a dual-species biofilm of *F.oxysporum* and *C. albicans* in vitro. The total biomass of the *F. oxysporum/C. albicans* mixed biofilm was lower compared to that of the single biofilms; instead, the metabolic activity of the biofilms was almost the same

The competition between the two species was highlighted in the formation of mixed biofilm. Our results showed the predominance of *C. albicans* in all the conditions tested, as shown by CFU count, likely due to the fact that *C. albicans* metabolism was faster, generating competition for adhesion sites.

Furthermore, the highest viable biomass detected where *C. albicans* was added after 48 h from *F. oxysporum* could be due to the development of *Candida* biofilm once *Fusarium* was allowed to adhere, develop, and mature as biofilm, forming a substrate for *Candida* cells.

Indeed, in mixed biofilms, interactions between species during their development are influenced by the sharing of nutrients available, so interaction is modulated by the environment and the coordination of intercellular communications.

Due to the increase in drug resistance among the most common pathogenic fungi against the conventional antifungal drugs, the combined treatment strategy is becoming widespread and may result in more effective treatment. Here, we demonstrated that the combination of WMR and FLC had a strong synergistic effect on the eradication of mature mixed biofilms that had been formed with *C. albicans* added after 48 h from *F. oxysporum*. The concentrations of 12.5 µM WMR and 30 µM FLC resulted in 60% biofilm eradication, indicating that the two drugs’ combination exerts broad prospects for the treatment of this biofilm-related infection.

These results have been confirmed by qRT-PCR, where four of the major transcriptional genes that govern biofilm formation, such as *ALS3* and *ERG11* genes for *C. albicans* and *VeA* and *VeLB* for *F. oxysporum*, showed significant downregulation of the related expression. The downregulation of *ALS3*, which is associated with adhesion in *C. albicans*, supported our results of in vitro biofilm eradication. On the contrary, WMR/FLC treatment increased expression levels of *ERG11*, indicating a possible negative interaction of the combination targeting enzyme CYP51 encoded by *ERG11* and leading to azole resistance, but it needs to be pointed out that *ERG11* is only one of the genes of the ergosterol biosynthesis pathway (*ERG1*, *ERG2*, *ERG3*, and *ERG11*) [38]. *F. oxysporum VeA* and *VelB* genes, encoding for key regulators of the velvet complex, influence the growth and development of fungal hyphae, conidiation, secondary metabolism, and pathogenicity in both plants and mammals [39]. Importantly, the expression of both *VeA* and *VelB* genes was downregulated, showing that the synergistic combination that counteracted *F. oxysporum* in dual-species biofilms exerts effects on these specific virulent factors.

As in other relevant studies [40], we opted to use the *G. mellonella* model to validate the in vivo performance of the combination treatment since it is a model that provides a rapid, inexpensive, and reliable way to evaluate the effects and toxicity in vivo of drugs. Our data indicated that treatment with WMR/FLC prolonged and increased the survival of *G. mellonella* larvae infected with the two microorganisms up to 80%. This finding was corroborated by the histological analyses of infected larvae treated with the synergistic combination of WMR/FLC, which showed a reduction in tissue damage and suggested the excellent synergy of the compounds in vivo.

In conclusion, the present work confirms that the combination treatment of WMR/FLC is able to eradicate the mixed biofilm in vitro and attenuates infection in the *G. mellonella* model. This work reveals the in vivo therapeutic potential of antimicrobial peptides combined with conventional drugs for controlling fungal infections and lays the foundations for future studies based on new formulations to fight against drug resistance.

## Figures and Tables

**Figure 1 pharmaceutics-14-01167-f001:**
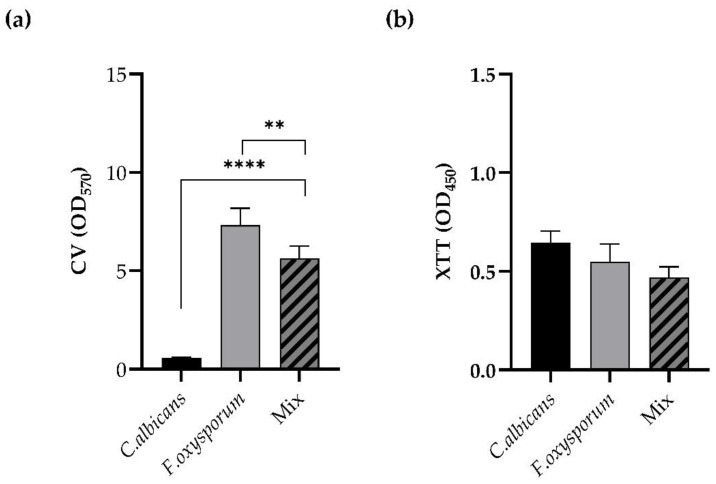
Panel (**a**): Comparison between the total biomass of monospecies biofilms of *C. albicans* and *F. oxysporum* with the dual-species biofilms formed by *C. albicans/**F. oxysporum* starting from a ratio of 1:1. Panel (**b**): Comparison between the metabolic activities of monospecies biofilms of *C. albicans* and *F. oxysporum* and the dual-species *C. albicans/**F. oxysporum* biofilm formed starting from a ratio of 1:1. Statistical significance: ** *p* < 0.01; **** *p* < 0.0001 (Tukey’s test).

**Figure 2 pharmaceutics-14-01167-f002:**
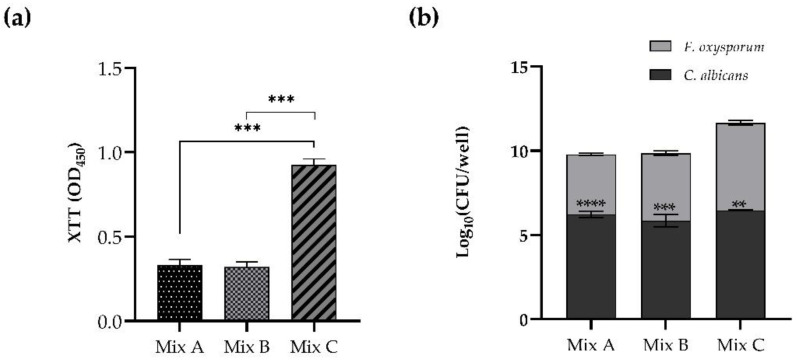
Panel (**a**). Metabolic activity of mixed biofilm. *F. oxysporum* inocula were dispensed, as first colonizer, allowing adherence (4 h, Mix A), formation (24 h, Mix B), and maturation (48 h, Mix C) before *C. albicans* addition h. Panel (**b**). Competitive behavior during formation of mixed biofilm. Log of colony-forming units (CFU) per well isolated from dual-species biofilms formed in the three different conditions of Mix A, Mix B, and MixC. Significant differences were determined by Tukey’s test, ** *p* < 0.01; *** *p* < 0.001, **** *p* < 0.0001.

**Figure 3 pharmaceutics-14-01167-f003:**
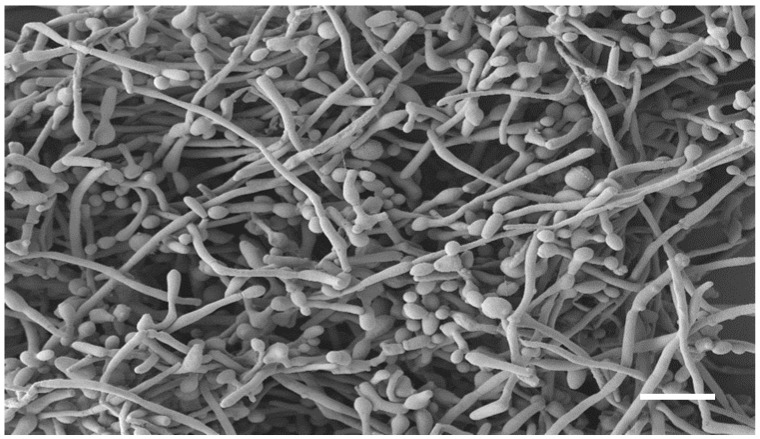
SEM image of the dual-species biofilm of *F. oxysporum* and *C. albicans*. Bar corresponds to 10 μm.

**Figure 4 pharmaceutics-14-01167-f004:**
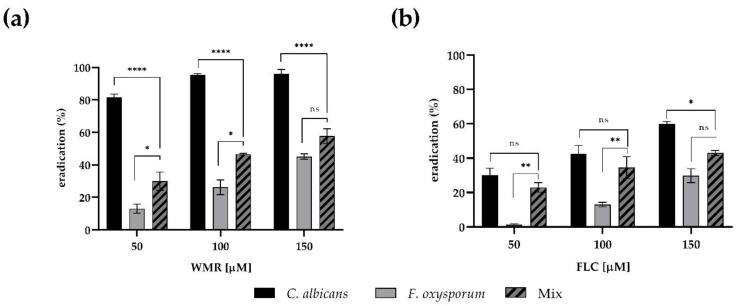
Eradication of WMR (panel **a**) and FLC (panel **b**) on mono- and dual-species biofilms formed on microplate surfaces quantified using the CV method. Statistical significance: * *p* < 0.05; ** *p* < 0.01; **** *p* < 0.0001; ns = not significant (Tukey’s test).

**Figure 5 pharmaceutics-14-01167-f005:**
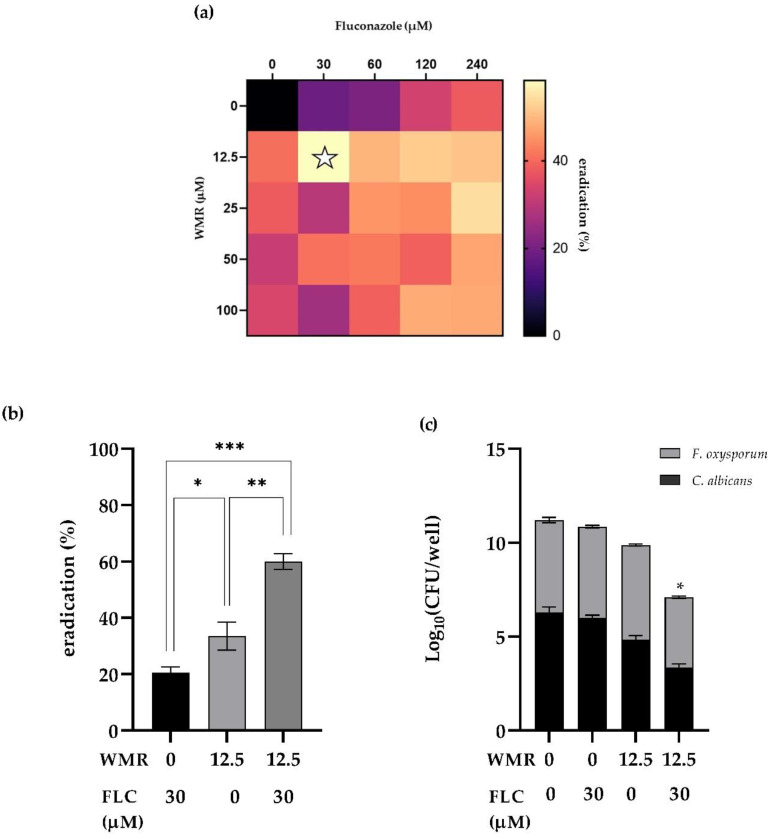
Assessment of antibiofilm activity of WMR and FLC against pre-formed dual-species biofilms formed by *C. albicans* and *F. oxysporum* after 24 h treatment. (**a**) Quantification of biofilm biomass by crystal violet staining after treatment of combinatorial agents. White star indicates the selected combinatorial concentrations. (**b**) Comparative analysis of individual and combinatorial efficacy of WMR and FLC at synergistic combination concentrations against dual-biofilm of *C. albicans*/*F. oxysporum.* (**c**) Bar graph representing the number of CFU by *C. albicans* and *F. oxysporum* grown in the absence and presence of identified synergistic combination. Statistical significance: * *p* < 0.05; ** *p* < 0.01; *** *p* < 0.001; ns = not significant (Tukey’s test).

**Figure 6 pharmaceutics-14-01167-f006:**
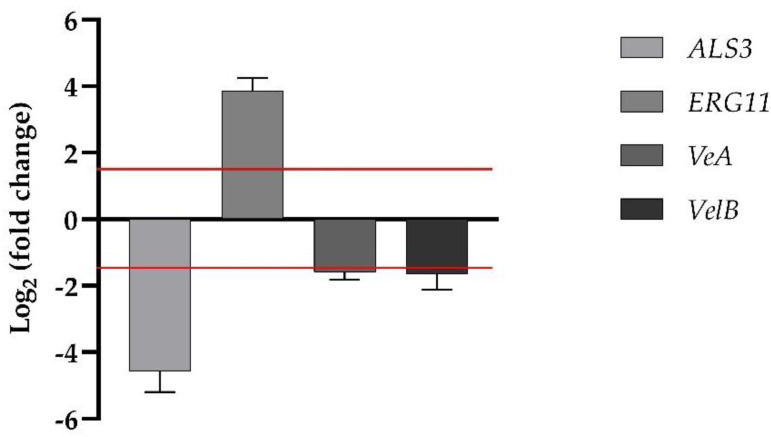
Real-time qPCR after treatment with compounds at synergistic concentrations. Histograms show the differences in the expression levels of four genes; fold differences greater than ±1.5 (see dotted red horizontal guidelines at values of +1.5 and −1.5) were considered significant.

**Figure 7 pharmaceutics-14-01167-f007:**
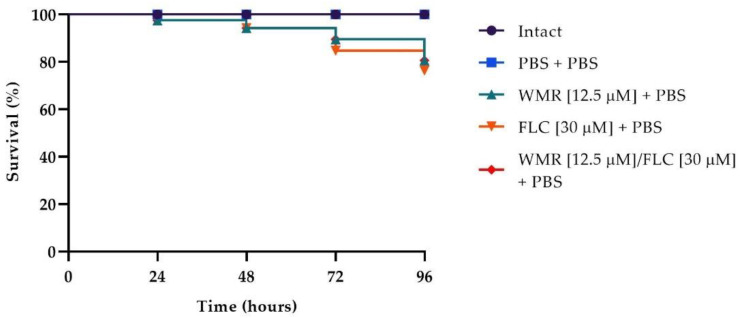
Toxicity of WMR and FLC and their combination on *G. mellonella* larvae.

**Figure 8 pharmaceutics-14-01167-f008:**
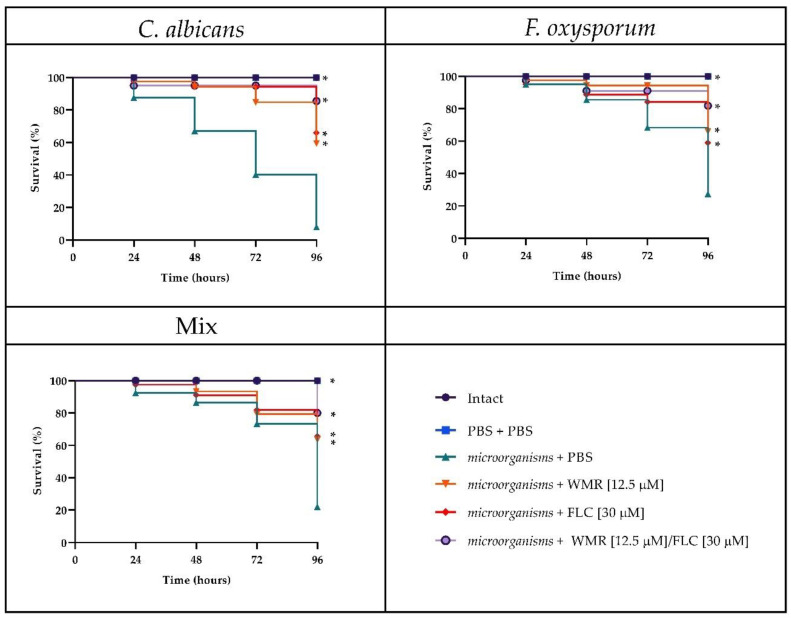
Survival curve of *G. mellonella* infected with *C. albicans*, *F. oxysporum*, and the mix of the two pathogens, treated with WMR and FLC and their combination. * indicates significant difference (*p* < 0.05, Tukey’s test).

**Figure 9 pharmaceutics-14-01167-f009:**
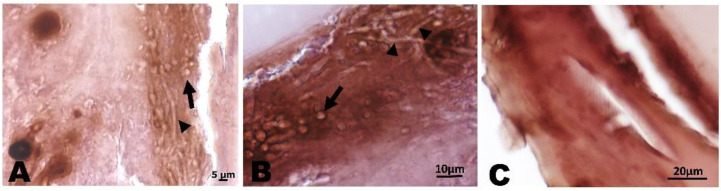
*G. mellonella* histopathology stained with hematoxylin and eosin. (**A**,**B**): *C. albicans* cells (arrow) and *fungal* hyphae (arrowhead) in infected *G. mellonella* larvae. (**C**): G. *mellonella* larvae treated with WMR peptide in combination with FLC.

**Table 1 pharmaceutics-14-01167-t001:** Susceptibility of the two fungal species to WMR and FLC.

	WMR (µM)	FLC (µM)
*C. albicans* ATCC 90028	150	>150
*F. oxysporum* NRRL 32931	>150	>150

## Data Availability

All data is contained within the article. For any additional information, please contact the corresponding author.

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
