# Peer review of "Competitiveness during Dual-Species Biofilm Formation of Fusarium oxysporum and Candida albicans and a Novel Treatment Strategy"

_pharmaceutics, 2022, doi:10.3390/pharmaceutics14061167_

Round 1

Reviewer 1 Report

The authors described both the ability of a Fusarium oxysporum isolate, NRRL 105 32931 and the Candida albicans strain ATCC 90028 to form a mixed biofilm, as well as the effects of the antimicrobial peptide WMR alone and in association with fluconazole (FLC) on mature mixed biofilm. In addition, the effectiveness of WMR and FLC alone and in combination on survival and tissue invasion was investigated in an alternative model with Galleria mellonella larvae infected with F. oxysporum and C. albicans. It was observed that C. albicans antagonized the growth of F. oxysporum in biofilm and the synergistic effect of the association WMR and FLC in in vitro assays and in the model with G. mellonella. The manuscript is sound and the conclusions are supported by the findings. However, the text should be revised, as some parts need language improvements, before it can be accepted. In addition, there is a need to organize the presentation of the results and the discussion. Below are some comments:

Line 18: Please, the authors should use “are” instead of “is”;

Line 125: Please, in the sentence “…incubated at of 28 °C…” delete “of”;

Line 156: add a comma after “Briefly”;

Line 175: “Stepanović” is misspelled (Stepanovi ’c);

Line 187-193: Please authors should clarify the procedure, mainly because the counts were very high in the first bars with log10 (CFU/well) > 10 (Figure 5c). Was biofilm formation assay performed in 24- or 96-well plates? How were microorganisms removed to prepare the suspensions? How far were the decimal dilutions prepared?

Line 266: Please, the authors should spell “Candida” in italics.

Line 294-299: The authors begin the presentation of the results by the susceptibility of the microorganisms, but the order is not logical. I suggest that this paragraph be moved to after line 352;

After figure 2, figure 4 is shown (figure 3 is not included). Please renumber the figures.

Line 417: “survival” is misspelled;

The discussion section is a bit confusing. In the 2nd paragraph (a short sentence from line 443 – 444), the discussion on combination of antifungal drugs and peptide activity is initiated. From lines 463 to 483, the text talks about biofilm. From line 484-491, the authors go back to talking about the activity of the peptide and then the text remains confused and does not flow in a logical way. Therefore, I suggest that the discussion be rewritten to make it more suitable.

Author Response

Response to Reviewer 1 Comments

The authors described both the ability of a Fusarium oxysporum isolate, NRRL 105 32931 and the Candida albicans strain ATCC 90028 to form a mixed biofilm, as well as the effects of the antimicrobial peptide WMR alone and in association with fluconazole (FLC) on mature mixed biofilm. In addition, the effectiveness of WMR and FLC alone and in combination on survival and tissue invasion was investigated in an alternative model with Galleria mellonella larvae infected with F. oxysporum and C. albicans. It was observed that C. albicans antagonized the growth of F. oxysporum in biofilm and the synergistic effect of the association WMR and FLC in in vitro assays and in the model with G. mellonella. The manuscript is sound and the conclusions are supported by the findings. However, the text should be revised, as some parts need language improvements, before it can be accepted. In addition, there is a need to organize the presentation of the results and the discussion. Below are some comments:

Line 18: Please, the authors should use “are” instead of “is”;

Line 125: Please, in the sentence “…incubated at of 28 °C…” delete “of”;

Line 156: add a comma after “Briefly”;

Line 175: “Stepanović” is misspelled (Stepanovi ’c);

We thank the referee for suggestions and we modified the MS accordingly.

Line 187-193: Please authors should clarify the procedure, mainly because the counts were very high in the first bars with log10 (CFU/well) > 10 (Figure 5c). Was biofilm formation assay performed in 24- or 96-well plates? How were microorganisms removed to prepare the suspensions? How far were the decimal dilutions prepared?

We thank the referee for his/her comments and indeed biofilm formation assay was performed in 96-well plates. After 48 h of incubation, the plates were emptied and washed with PBS to remove non-adherent cells and 100 μL of PBS was added in each well. Adherent cells were accurately scraped from wells, vortexed and suspended in 1 mL of PBS. Decimal dilutions were prepared up to 10-5, and 100 μL suspension was plated on selective media (187-193).

Line 266: Please, the authors should spell “Candida” in italics.

We thank the referee for his suggestion and we modified the MS accordingly

Line 294-299: The authors begin the presentation of the results by the susceptibility of the microorganisms, but the order is not logical. I suggest that this paragraph be moved to after line 352;

We thank the referee for his comment. We think it is better to show  the susceptibility of F. oxysporum and C. albicans planktonic cells to the examined antifungal agents as shown in Table 1, and then test the possible formation of  mixed biofilm and WMR/FLC  eradication capability

After figure 2, figure 4 is shown (figure 3 is not included). Please renumber the figures.

 We thank the referee for his suggestion and we modified the MS accordingly

Line 417: “survival” is misspelled;

We thank the referee for his suggestion and we modified the MS accordingly

The discussion section is a bit confusing. In the 2nd paragraph (a short sentence from line 443 – 444), the discussion on combination of antifungal drugs and peptide activity is initiated. From lines 463 to 483, the text talks about biofilm. From line 484-491, the authors go back to talking about the activity of the peptide and then the text remains confused and does not flow in a logical way. Therefore, I suggest that the discussion be rewritten to make it more suitable.

We thank the referee for his comment . We have re-written the discussion more clearly in the revised MS .

Reviewer 2 Report

The manuscript “Competitiveness during dual-species biofilm formation of Fusarium oxysporum and Candida albicans and a novel treatment strategy” is well written and easy to read. They presented an in-depth analytical study of the WMR peptide and fluconazole effect on biofilms formation of two fungi, including in vivo studies on Galleria mellonella larvae as a model. However, I have a few observations to improve it:

Line 24: Please indicate the meaning of WMR, it is the first time it is mentioned.

Line 123-124: Please describe the complete methodology to wash and obtain Candida cells, if you centrifugated what were the centrifugation speed, temperature, time, etc.

Line 157: What is the meaning of RPMI?

Line 314-315: You mention “Metabolic activities were similar, with a slight highest value in the case of Candida monospecies biofilm” please do a statistical analysis to support o refuse this statement.

Figure 7 and 8: What do you mean "intact"? please be clearer.

Line 496: Please write C. albicans in italics. 

Author Response

Response to Reviewer 2 Comments

The manuscript “Competitiveness during dual-species biofilm formation of Fusarium oxysporum and Candida albicans and a novel treatment strategy” is well written and easy to read. They presented an in-depth analytical study of the WMR peptide and fluconazole effect on biofilms formation of two fungi, including in vivo studies on Galleria mellonella larvae as a model. However, I have a few observations to improve it:

Line 24: Please indicate the meaning of WMR, it is the first time it is mentioned.

We thank the referee for his/her comments and indeed the antimicrobial  peptide WMR (14-amino-acid-peptide: NH2-WGIRRILKYGKRSK-CONH2) has been previously identified in our lab as a modification of the native sequence of myxinidin.In the revised manuscript the phrase “” a modified version of the native peptide myxidin, has been also introduce in the Abstract 

Line 123-124: Please describe the complete methodology to wash and obtain Candida cells, if you centrifugated what were the centrifugation speed, temperature, time, etc.

We thank the referee for his/her comments and indeed C. albicans cells were maintained on Sabouraud dextrose agar and cultured in Tryptone Soy Broth supplemented with 1% glucose for 16–18 h at 37 °C. Then cultures were centrifuged 5000 rpm for 3 min at 4 °C and washed twice with PBS.

Line 157: What is the meaning of RPMI?  

RPMI means, Roswell Park Memorial Institute;but it is generally reported as RPMI

Line 314-315: You mention “Metabolic activities were similar, with a slight highest value in the case of Candida monospecies biofilm” please do a statistical analysis to support o refuse this statement.

We have re-written the sentence.

Figure 7 and 8: What do you mean "intact"? please be clearer.

. The term intact refers to untreated larvae.

Line 496: Please write C. albicans in italics. 

Done

Reviewer 3 Report

Dear Authors,

The submitted manuscript is focused on mixed biofilms formed by two fungal species, which constitute a serious problem in therapy. In the introduction, the background and importance of this research is well explained. The design of the article and selected methods are appropriately described. 

Comments and recommendations:

I recommend adding also a note about fluconazole concentration in μg/ml regarding the breakpoints of CLSI to the material and methods. I understand that for comparison with peptide, authors selected usage of "μM". In general, for easier comparison of these obtained susceptibility results of fluconazole to other publications, I suppose to use both μM (μg/ml).

Line 171: polystirene – check grammar, please

Line 262: 12,5 μM WMR  - there is comma instead of dot

"Figure 2". SEM image of the dual-species biofilm of F. oxysporium and C. albicans. – Rename, please, as a "Figure 3" and correct, please name of F. oxysporum

My question for authors is: How do you know to distinguish F. oxysporum hyphae and hyphae of C. albicans in the SEM image or in the figure of larvae histopathology? 

Line 496 C. albicans, in vitro – missing italics

Author Response

Response to Reviewer 3 Comments

Dear Authors

The submitted manuscript is focused on mixed biofilms formed by two fungal species, which constitute a serious problem in therapy. In the introduction, the background and importance of this research is well explained. The design of the article and selected methods are appropriately described. 

Comments and recommendations:

I recommend adding also a note about fluconazole concentration in μg/ml regarding the breakpoints of CLSI to the material and methods. I understand that for comparison with peptide, authors selected usage of "μM". In general, for easier comparison of these obtained susceptibility results of fluconazole to other publications, I suppose to use both μM (μg/ml).

We thank the referee for his suggestion and we modified the MS accordingly. Fluconazole concentrations in μg/ml have been reported in the revised text

Line 171: polystirene – check grammar, please

We have checked.

Line 262: 12,5 μM WMR  - there is comma instead of dot

We have checked.

"Figure 2". SEM image of the dual-species biofilm of F. oxysporium and C. albicans. – Rename, please, as a "Figure 3" and correct, please name of F. oxysporum 

Done

My question for authors is: How do you know to distinguish F. oxysporum hyphae and hyphae of C. albicans in the SEM image or in the figure of larvae histopathology? 

We thank the referee for his suggestion and we modified the MS accordingly. In the revised version we report the generic term “fungal” hyphae.

Line 496 C. albicans, in vitro – missing italics

We have checked

Reviewer 4 Report

This manuscript, entitled “Competitiveness during dual-species biofilm formation of Fusarium oxysporum and Candida albicans and a novel treatment strategy”, had described the interactions between C. albicans and F. oxysporum dual-species biofilm, and the ability of the WMR peptide with/without fluconazole to eradicate the mixed biofilm. The study is of interest, however, before further evaluation, I would like to confirm about the background and aim of this study.

Firstly, in the introduction, Fusariosis has been mentioned as an invasive fungal infection caused by Fusarium spp.. However, as far as I know, C. albicans is not a major species isolated from skin. What’s the chance of co-exist of Fusarium oxysporum and Candida albicans? Has C. albicans been isolated from Fusariosis samples. Please clarify in the introduction.

Secondly, suppose C. albicans been isolated from Fusariosis samples and co-exist with Fusarium oxysporum. Fusarium oxysporum should be the dominant species with much higher cell numbers than C. albicans. However, the dual-species biofilm model set up in this study is 1:1 C. albicans and F. oxysporum. Thus, I doubt the model to reflect real situation.

Thirdly, proper dual-species model reflecting the real situation and the interaction between Fusarium oxysporum and Candida albicans would be interesting. In depth mechanism could be studied. However, such study is not within the scope of Pharmaceuticals.

Fourthly, the antifungal and antibiofilm effect of WMR peptide on albicans and non-albicans Candida species, as well as the effect of peptide WMR-K on dual-species biofilm Candida albicans/Klebsiella pneumoniae have been reported. The antifungal and antibiofilm effect of WMR peptide on C. albicans have been proved. This study yielded the same result, which lacking novelty. The WMR peptide part is the most suitable part to Pharmaceuticals. However, no significant findings were shown.

Author Response

Response to Reviewer 4 Comments

This manuscript, entitled “Competitiveness during dual-species biofilm formation of Fusarium oxysporum and Candida albicans and a novel treatment strategy”, had described the interactions between C. albicans and F. oxysporum dual-species biofilm, and the ability of the WMR peptide with/without fluconazole to eradicate the mixed biofilm. The study is of interest, however, before further evaluation, I would like to confirm about the background and aim of this study.

Firstly, in the introduction, Fusariosis has been mentioned as an invasive fungal infection caused by Fusarium spp.. However, as far as I know, C. albicans is not a major species isolated from skin. What’s the chance of co-exist of Fusarium oxysporum and Candida albicans? Has C. albicans been isolated from Fusariosis samples. Please clarify in the introduction.

We thank the referee for his comment. We revised the introduction accordingly and expanded  information about Fusarium and invasive fusariosis.Further, the possible occurrence of Fusarium/Candida biofilm in vivo has been clarified in the revised text indeed as reported from other researchers (Jenks, J.D.; Reed, S.L.; Seidel, D.; Koehler, P.; Cornely, O.A.; Mehta, S.R.; Hoenigl, M. Rare mould infections caused by Mucorales, Lomentospora prolificans and Fusarium, in San Diego, CA: The role of antifungal combination therapy. Int. J. Antimicrob. Agents 2018, 52, 706–712.)  Candida albicans is the most prevalent species implicated in both invasive and mucocutaneous infections. Nowadays invasive candidiasis is a serious infection in immunocompromised patients as well as the increase of invasive mould infections with non-Aspergillus species. Indeed, recently, Fusarium species, which are more often resistant to voriconazole and posaconazole, have also been observed in immunosuppressed hosts. (lines 112-119)

Secondly, suppose C. albicans been isolated from Fusariosis samples and co-exist with Fusarium oxysporum. Fusarium oxysporum should be the dominant species with much higher cell numbers than C. albicans. However, the dual-species biofilm model set up in this study is 1:1 C. albicans and F. oxysporum. Thus, I doubt the model to reflect real situation.

We thank the referee for his comment. As reported (Kontoyiannis, D.P.; Marr, K.A.; Park, B.J.; Alexander, B.D.; Anaissie, E.J.; Walsh, T.J.; Ito, J.; Andes, D.R.; Baddley, J.W.; Brown, J.M.; et al. Prospective Surveillance for Invasive Fungal Infections in HematopoieticStem Cell Transplant Recipients, 2001–2006: Overview of the Transplant-Associated Infection Surveillance Network (TRANSNET) Database. Clin. Infect. Dis. 2010, 50, 1091–1100.) immunotherapies have revolutionized the treatment of tumors and autoimmune diseases but have highlighted infectious risks that are only now beginning to be considered especially the infectious risk for opportunistic fungal infections. Both these microorganisms, Candida albicans and Fusarium oxysporum, are able to form biofilms on biotic and abiotic surfaces like catheters, contact lents, prosthetic valves, and can be  transmitted through physical touch to neonates in the hemodialysis systems used by chronic renal patients, thereafter evolving into bloodstream infections. The formation of mixed biofilms between these fungi is likely since biofilms provide many advantages for the microorganisms involved and rarely involve only one species.

So long as the heterogeneity of species within biofilms makes it difficult characterize the contribution of each microorganism to pathogenesis and maintenance of infection and disease. Our aim has been  to study the competition between the two species by simulating various conditions in vitro since in mixed biofilms, interactions between different species can be synergistic, neutral or antagonist, it is necessary to evaluate different conditions of biofilm formation also to evaluate their susceptibility to new antifungal drugs.

Thirdly, proper dual-species model reflecting the real situation and the interaction between Fusarium oxysporum and Candida albicans would be interesting. In depth mechanism could be studied. However, such study is not within the scope of Pharmaceuticals.

We thank the referee for his suggestion.  Our future investigations will certainly be directed to the study of the biofilm eradication mechanism even if we can hypothesize a disturbance in the biofilm life-style maybe  involving the sharing of nutrients available or the intercellular communications that take place through quorum sensing molecules, which coordinate the collective behavior of microorganisms in these community,  due to a perturbing action of the two drugs.

Fourthly, the antifungal and antibiofilm effect of WMR peptide on albicans and non-albicans Candida species, as well as the effect of peptide WMR-K on dual-species biofilm Candida albicans/Klebsiella pneumoniae have been reported. The antifungal and antibiofilm effect of WMR peptide on C. albicans have been proved. This study yielded the same result, which lacking novelty. The WMR peptide part is the most suitable part to Pharmaceuticals. However, no significant findings were shown.

We thank the referee for his comment. Following the invitation,, based on our expertise, to contribute to the Special Issue: “Recent Advances in Prevention and Treatment of Eukaryotic Infectious Diseases Website: https://www.mdpi.com/journal/pharmaceutics/special_issues/eukaryotic_infection” we think that in the presented paper significant and novel results have been obtained about the WMR peptide synthetized in our laboratory that it has never been tested against filamentous fungi

Round 2

Reviewer 4 Report

The revision has been improved according to the comments and the response has answered and explained some of my concerns. However, the authors failed to address some points.

Firstly, as I stated in my previous comment, the antifungal and antibiofilm effect of WMR peptide on albicans and non-albicans Candida species, as well as the effect of peptide WMR-K on dual-species biofilm Candida albicans/Klebsiella pneumoniae have been reported. The antifungal and antibiofilm effect of WMR peptide on C. albicans have been proved. This study yielded the same result, which lacking novelty. The WMR peptide part is the most suitable part to Pharmaceuticals. However, no significant findings were shown.

In the author’s response, the authors state “we think that in the presented paper significant and novel results have been obtained about the WMR peptide synthetized in our laboratory that it has never been tested against filamentous fungi”. Please check the following link about the published papers about the antifungal and antibiofilm effect of WMR peptide on Candida albicans.

https://pubmed.ncbi.nlm.nih.gov/33669279/

https://pubmed.ncbi.nlm.nih.gov/35216270/

Secondly, in Candida albicans, ECE1 gene has been reported to contribute to its pathogenesis and cause candidiasis. Also, the variation in ECE1 gene sequences contribute to its reduced pathogenicity of Candida albicans in candidiasis. It would be interesting to see the change in ECE1 expression and the difference in ECE1 sequences upon the treatment of WMR peptide in this study.